# Impulse oscillometry system for assessing small airway dysfunction in pediatric bronchiolitis obliterans; association with conventional pulmonary function tests

Hye Jin Lee[1]*, Hwan Soo Kim[2], Jong-seo Yoon[3]

1 Department of Pediatrics, Seoul St. Mary's Hospital, College of Medicine, The Catholic University of Korea, Seoul, South Korea, 2 Department of Pediatrics, Bucheon St. Mary's Hospital, College of Medicine, The Catholic University of Korea, Seoul, South Korea, 3 Department of Pediatrics, Eunpyeong St. Mary's Hospital, College of Medicine, The Catholic University of Korea, Seoul, South Korea

* pedhjlee@gmail.com

## Abstract

Impulse oscillometry system (IOS) is a simple, and less invasive method for assessing small to total airway resistance in children. We analyzed the correlation between IOS, spirometry, and plethysmographic parameters performed for the diagnosis of pediatric BO patients. A total of 89 IOS assessments of pediatric BO patients or children without lung disease were included, and the relationship between pulmonary function tests (PFTs) and diagnostic performance was analyzed. R5, R5-20, X5, and AX were statistically significantly worse in the BO group. In general linear correlation analysis, R5% (adjusted β [aβ], -0.59; p < 0.001) and AX % (aβ, -0.9; p < 0.001) showed the strongest correlation with conventional PFT parameters. R5% and AX % also showed the highest correlation with FEF25-75% (aβ, -0.48; p < 0.001 and aβ, -0.83; p < 0.001), and sRaw % (aβ, -0.73; p = 0.003 and aβ, -0.59; p = 0.008, respectively). Multivariate logistic regression analysis showed that R5 Z-score showed the highest ORs with FEV1 (OR = 3.94, p = 0.006), FEF25-75% (OR = 5.96, p = 0.005), and sRaw % (OR = 4.85, p = 0.022). Receiver operating curve analysis suggested AX % and R5% as the most optimal IOS parameters for BO diagnostic performance with the area under the curve of 0.915 and 0.882, respectively. In conclusion, R5 and AX are the parameters that can independently identify the severity of airway obstruction in pediatric BO patients without conventional lung function tests. IOS is an easy-to-perform, and reliable diagnostic method capable of detecting pathological obliteration of the small airways in children with BO.

## Introduction

Bronchiolitis obliterans (BO) is a fibrosing chronic obstructive lung disease that leads to the obliteration of the small airways [1]. BO occurs secondarily to several conditions including infection, allogeneic hematopoietic stem cell transplantation (HSCT), and lung transplants [2–

**Funding:** The author(s) received no specific funding for this work.

**Competing interests:** The authors have declared that no competing interests exist.

4]. Pulmonary function tests (PFTs) are essential for the diagnosis and management of BO, and spirometry is considered the gold standard for determining the severity of the obstructive pulmonary disease. However, the accurate assessment of the degree of airway obliteration in young children with BO is challenging due to the relative inaccessibility of the small airways and the complexity of the distal lung lesions involved.

The impulse oscillometry system (IOS) is increasingly used to measure lung function, especially in young children because of their ability to use only normal tidal volume breathing rather than the effort and coordination required for spirometry. Compared to conventional spirometry, IOS is a simple, and less invasive method for assessing small to total airway resistance in children [5]. Recent studies have shown that IOS is capable of quantifying the degree of small airway impairments [6–8]. The correlation between IOS and conventional PFTs has already been evaluated in children with asthma [7, 9–12], cystic fibrosis [13, 14], and other chronic obstructive lung diseases [15, 16]. Previous studies of IOS have suggested airflow limitation in the small airways, distal airway function with bronchodilators, and response to corticosteroids [17]. To the best of our knowledge, studies of using IOS with pediatric BO patients are limited. In young children with BO, irreversible fibrosis and fixed small airway obliteration occur in a more severe form compared to other obstructive lung diseases. Therefore, spirometry requiring forced ventilation or the body box with long test times has a low success rate, particularly in severe cases. The purpose of this study is, first, to verify the correlation between IOS, spirometry, and body plethysmography in children with BO. Second, and more importantly, we tried to suggest that the IOS parameters are relatively easily obtainable, and reliable indicators even in patients with severe BO who cannot perform conventional PFTs.

## Subjects and methods

### Study population

We conducted a retrospective chart review of pediatric patients under 18 years of age who underwent IOS from March 2018 to February 2019 at the Department of Pediatrics, Seoul St. Mary's Hospital, College of Medicine at the Catholic University of Korea. Inclusion criteria were (a) physician-diagnosed BO (n = 26), patients who underwent IOS with (b) pre-transplant evaluation in the Department of hemato-oncology without previous lung disease (n = 14), and (c) healthy child (n = 1); Both (b) and (c) were included in the control group. Patients with asthma, chronic lung disease of prematurity, or obstructive lung disease other than BO were excluded from the study (n = 13). IOS, spirometry, and/or body plethysmography were repeated at outpatient visits with written informed consent obtained from the parent or guardian of each participant. This study was approved by the Institutional Review Board (IRB) Committee of Seoul Saint Mary's hospital (IRB number: KC21RISI0028).

### IOS and PFTs

All IOS procedures were performed using Tremoflo (Thorasys Inc., Montreal, QC) with measurements made according to the standards adopted by the European Respiratory Society [18]. IOS was performed with the child sitting upright, a clip to block the nose, lips sealed around the mouthpiece, and cheeks supported with the hands to decrease dead space. The mean values of at least two recordings that met the criteria of > 0.80 coherence were used for analysis. The IOS parameters, Resistance at 5 Hz (R5), the difference between R5 and R20 Hz (resistance of small-to-mid-sized airways, R5-20), reactance at 5 Hz (reactance of more central, X5), and area of reactance (AX) were measured, and they were interpreted as % predictive value and Z-score by reference, Calogero et al. 2013 [19]. Other PFTs were performed with a Vmax

instrument (Sensor Medics, VIASYS Healthcare, Yorba Linda, CA) according to guidelines from the American Thoracic Society and European Respiratory Society.

## Statistical analysis

Descriptive analyses of qualitative variables are expressed as the number of patients for each category and percentage. Quantitative variables are presented as mean and standard deviation for normally distributed variables or as median and interquartile range (IQR) when not. Comparisons between patients with BO and the control group were done using the Chi-square test for categorical variables and the Mann-Whitney test for continuous variables. If there is at least one cell that has a frequency of five or less, an additional correction was done with Fisher's exact test. The IOS data were square-root transformed to obtain a normal distribution, and a general linear model analysis between the conventional PFTs was presented with beta coefficients with 95% confidence intervals (CIs). Results of the multivariate logistic regression with obstructive PFT parameters were presented as odds ratios (OR) with 95% CI. All multivariate outcomes were adjusted for confounding by age and sex (based on the assumption that these are potential confounders). The receiver-operating characteristic (ROC) method was used to evaluate the accuracy of the IOS parameters for predicting physician-diagnosed BO. Areas under the curve (AUC) and optimal cut-off points based on maximizing the sum of the sensitivity and specificity were calculated for each IOS parameter. All p-values of $< 0.05$ were considered statistically significant unless otherwise stated. All statistical analyses were performed using R Version 4.0.5.

## Results

### Patient characteristics

The demographic and clinical characteristics of the patients are presented in Table 1. A total of 89 IOSs were performed in 41 participants, including 15 controls and 26 BO patients. Spirometry and body plethysmography were simultaneously performed in 43 (48.3%) and 29 (32.6%) cases, and there were cases of failure in patients who were too young or unable to take forced inspiration and expiration due to severe BO.

Of the 26 patients in the BO group, 20 (76.9%) had previous HSCT, 5 (19.2%) with PIBO, and 1 (3.8%) with SJS patients were included. The median ages of the control and BO groups were 7.0 and 12.5 years, respectively. In the PFTs, FVC, FEV1, FEV1/FVC, FEF25-75%, and RV were statistically significantly worse in the BO group. Raw% and sRaw% were significantly higher in patients with BO. Most of the IOS parameters, R5-Z, R5%, R5-20, X5, AX cmH2O/L, AX-Z, AX % were statistically significantly worse in the BO group (Table 1, Fig 1).

### General linear and logistic regression analysis

Our results showed that IOS parameters, the R5, R5-20, and AX showed a significant association with conventional airway obstruction and plethysmographic resistance parameters (Fig 2). The multivariate linear regression analysis showed that IOS parameters had a statistically significant correlation with FEV1, FEF25-75%, and sRaw % (Table 2). Particularly, the R5% and AX % parameters showed the highest relative correlation with PFTs. Fig 3 illustrates the relative prediction weights of the IOS parameters for obstructive PFTs using backward elimination analysis. The combination and importance of the IOS variables configured according to the PFT parameters appeared different, but AX was the only IOS parameter with the highest relative importance for all obstructive PFT values.

**Table 1. Patient characteristics in all patients.**

| Patients (N/cases) | Control | BO | Total | *P*-value |
|---|---|---|---|---|
| | (N = 15/17) | (N = 26/72) | (N = 41/89) | |
| Median age (IQR), years | 7.0 [6.0; 10.0] | 12.5 [6.5;18.0] | 9.0 [6.0;18.0] | 0.011 |
| Gender, male | 10 (66.7%) | 20 (76.9%) | 30 (73.2%) | 0.728 |
| Diagnosis | | | | < 0.001 |
| BO-HSCT | - | 20 (76.9%) | 20 (48.8%) | |
| PIBO | - | 5 (19.2%) | 5 (12.2%) | |
| SJS | - | 1 (3.8%) | 1 (2.4%) | |
| R5, cmH2O/L/s | 8.6 [6.0; 9.5] | 8.2 [4.7;10.5] | 8.2 [5.5;10.2] | 0.904 |
| R5, Z-score | 0.4 ± 1.1 | 1.8 ± 1.4 | 1.6 ± 1.5 | < 0.001 |
| R5, % | 116.6 [97.7;133.9] | 154.0 [126.0;218.8] | 146.1 [119.8;193.8] | < 0.001 |
| R5-20, cmH2O/L/s | 1.1 ± 1.4 | 3.0 ± 2.1 | 2.7 ± 2.1 | < 0.001 |
| X5, cmH2O/L/s | -0.4 [-0.8; 0.4] | 1.2 [-0.4; 3.2] | 0.6 [-0.5; 2.3] | 0.001 |
| AX, cmH2O/L | 24.1 [16.7;40.1] | 72.5 [32.2;126.7] | 59.3 [24.1;99.8] | < 0.001 |
| AX, Z-score | 0.1 ± 1.2 | 2.4 ± 2.4 | 1.9 ± 2.4 | < 0.001 |
| AX, % | 120.3 [55.8;169.2] | 485.7 [144.6;2040.0] | 277.3 [111.0;1780.5] | < 0.001 |
| VT, L | 12.7 [9.0;16.9] | 6.1 [3.1;11.2] | 7.8 [3.8;12.7] | < 0.001 |
| COH5 | 0.9 [0.9; 0.9] | 0.9 [0.9; 0.9] | 0.9 [0.9; 0.9] | 0.975 |
| CV, R5 | 11.9 [8.7;17.1] | 8.0 [5.8;12.2] | 8.8 [5.9;14.1] | 0.012 |
| CV, AX | 34.1 [29.7;66.8] | 7.0 [3.9;15.3] | 8.7 [4.8;28.3] | < 0.001 |
| CV, VT | 12.7 [9.0;16.9] | 6.1 [3.1;11.2] | 7.8 [3.8;12.7] | < 0.001 |
| FVC % | 88.2 013.8; | 73.8 013.8; | 77.8 013.8; | 0.045 |
| FEV1% | 89.4 ± 14.4 | 50.9 ± 34.1 | 61.7 ± 34.5 | < 0.001 |
| FEV1/FVC % | 91.4 ± 5.9 | 60.5 ± 23.1 | 69.1 ± 24.2 | < 0.001 |
| FEF25-75% | 113.9 ± 33.9 | 43.3 ± 51.6 | 63.0 ± 56.8 | < 0.001 |
| TLC % | 97.1 ± 12.4 | 106.4 ± 23.7 | 103.2 ± 20.7 | 0.260 |
| RV % | 107.1 ± 45.7 | 183.4 ± 134.3 | 157.1 ± 116.7 | 0.034 |
| RV/TLC % | 27.9 ± 10.6 | 41.3 ± 16.5 | 36.3 ± 15.8 | 0.030 |
| Raw, cmH2O/L/s | 3.7 [2.8; 5.1] | 5.2 [3.6; 6.2] | 4.9 [3.0; 5.7] | 0.200 |
| Raw, % | 76.0 [60.0;91.5] | 156.0 [112.0;383.0] | 138.0 [68.0;242.0] | 0.010 |
| sRaw, cmH2O/L/s | 4.8 [3.7; 6.9] | 12.5 [5.4;31.6] | 7.6 [4.7;14.8] | 0.009 |
| sRaw, % | 105.0 [74.0;150.0] | 233.0 [109.0;701.0] | 150.0 [106.0;280.0] | 0.016 |

BO: bronchiolitis obliterans; HSCT: hematopoietic stem cell transplantation; IQR; interquartile range, PIBO: post infectious bronchiolitis obliterans; SJS: Swyer james syndrome.

Multivariate logistic regression analysis was performed to obtain the odds ratios (ORs) of the IOS for PFT and plethysmographic parameters by dividing into two groups; normal and obstructive (Table 3). The cutoffs for normal values used for FEV1, FEF25-75%, and sRaw were 75%, 60%, and 203.5% of predicted, respectively. The obstructive group was defined when one of these criteria was met. We used a cutoff for FEV1 < 75% based on the criteria for the definition of BO from the National Institutes of Health (NIH) consensus, 2014 [20]. FEF25-75% is a potentially sensitive parameter for evaluating small airway function in pediatrics [21]. There are no guidelines regarding normal values in children, and previous studies have suggested thresholds of 60% or 65% in children to define peripheral airflow obstruction [22, 23]. The normal reference ranges for sRaw also have not been established in children, so we used 203.5% as the cut-off value following our ROC analysis in this study. Most of the resistance and reactance IOS parameters showed statistically significant correlations with FEV1,

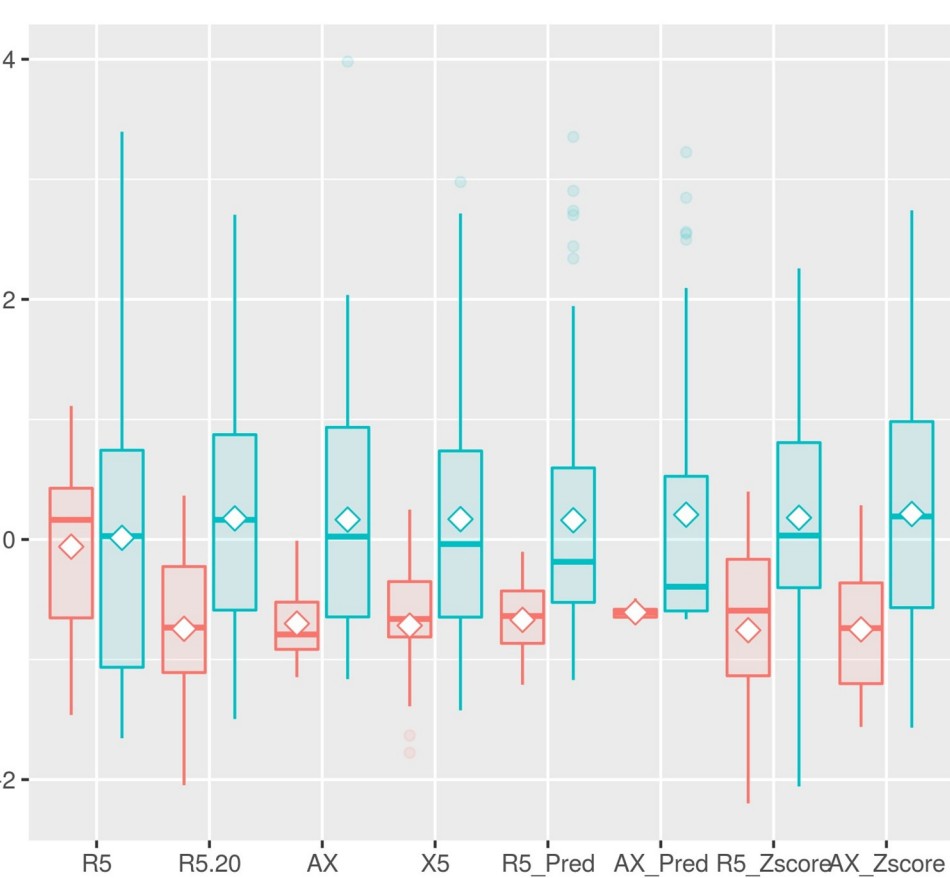

**Fig 1. Impulse oscillometry parameters in control and BO groups.** For better visualization, the Y-axis values have been rescaled for each unit of the IOS parameter.

FEV1/FVC, FEF25-75%, Raw, sRaw. The R5 Z-score showed the highest ORs with FEV1, FEF25-75%, and Raw (OR; 3.94, 5.96, and 4.85, p values; 0.006, 0.005, and 0.022, respectively).

## ROC analysis

In ROC analysis, the optimal cut-off values of each IOS and plethysmographic values for BO diagnosis were obtained, and the AUCs and sensitivity-specificity for each cut-off are presented in Table 4. The two most reliable IOS parameters with the most optimal diagnostic performance of BO, were AX % and R5% (AUC; 0.915 and 0.882, respectively).

## Discussion

IOS is a method of assessment based on forced vibration technology that has the advantage, particularly for children, of being relatively effort-independent [24, 25]. IOS can be used to measure respiratory resistance by applying small pressures to the mouth which are transmitted to the lungs. Our study showed that the proportion of children who performed IOS was much higher than that of spirometry or plethysmography. The IOS has enabled the assessment of lung function, especially in young children who have difficulty understanding and cooperating with conventional PFT.

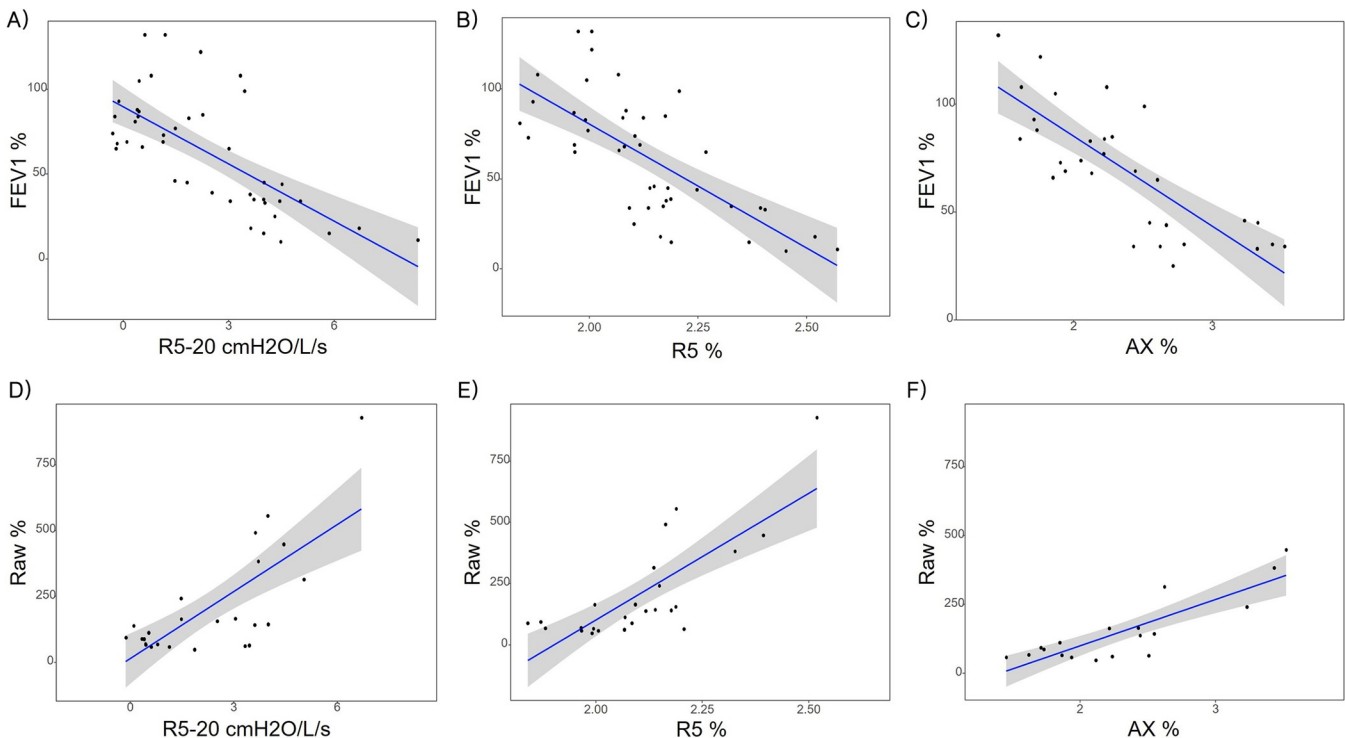

**Fig 2.** Linear regression plot between PFTs and log-transformed IOS parameters; A-C) Correlation with FEV1; (A) R5-20 cmH2O/L/s (B) R5% (C) AX %, D-F) correlation with Raw %; (D) R5-20 cmH2O/L/s (E) R5% (F) AX %.

In the BO group, most of the IOS parameters were significantly worse except raw R5 cmH2O/L/s. As there is no general reference range of IOS specific to pediatric patients to date, we assumed that validation should include raw values, Z-scores, and % predictive values with all different units. In the general linear regression analysis between IOS and PFT parameters, apparent associations were identified for R5%, and AX% with PFTs. These two parameters refer to resistance and reactance at different locations from the central to the peripheral airway. Resistance is a component of lung impedance and provides information about the forward pressure of the airway, and reactance is the repulsive resistance of distensible airways by area. Since the low oscillation frequencies of the IOS can be transmitted more distantly than the higher frequencies, resistance at 5Hz (R5) represents peripheral and total airway resistance, and resistance at 20 Hz (R20) represents the large airways [26–28]. It is well established that changes in resistance become evident at low frequencies in patients with peripheral airway disease [27]. Small airway resistance can be calculated by subtracting R20 from R5, and can be used with AX and X5 to reflect changes in the degree of obstruction in peripheral airways [28, 29].

Our results suggested that AX showed a distinct linear relationship with obstructive PFT variables, and was the most reliable parameter with a particularly high AUC value for BO diagnosis in ROC analysis. AX, a reactance parameter, represents the area under the curve which is the total reactance at all frequencies between 5 Hz and Fres [27]. AX, therefore reflects the elastic properties of the lungs in all frequency domains and also provides information on peripheral airway obstruction. Loss of compliance due to extensive fibrosis of peripheral airway tissue in BO patients can be better represented by this overall value than by resistance or reactance parameters at specific frequencies. There have been IOS studies of a range of diseases,

**Table 2. General linear regression analysis of airway resistance and reactance parameters between IOS, spirometry, and plethysmographic resistance.**

| | FEV1 | | | FEF25-75 | | | sRaw % | | |
|---|---|---|---|---|---|---|---|---|---|
| | std.Beta | 95% CI | | p | std.Beta | 95% CI | | p | std.Beta | 95% CI | | p |

| | std.Beta | 95% CI | | p | std.Beta | 95% CI | | p | std.Beta | 95% CI | | p |
|---|---|---|---|---|---|---|---|---|---|---|---|---|
| R5, cmH2O/L/s[†] | | | | | | | | | | | | |
| Crude | -0.26 | -0.56 | 0.05 | 0.098 | -0.60 | -0.86 | -0.35 | < .001 | 0.38 | -0.01 | 0.77 | 0.054 |
| Adjusted[‡] | -0.53 | -0.78 | -0.28 | < .001 | -0.44 | -0.69 | -0.18 | 0.001 | 0.39 | 0.11 | 0.68 | 0.008 |
| R5, Z-score | | | | | | | | | | | | |
| Crude | -0.59 | -0.85 | -0.34 | < .001 | -0.60 | -0.86 | -0.35 | < .001 | 0.49 | 0.12 | 0.85 | 0.012 |
| Adjusted[‡] | -0.47 | -0.73 | -0.21 | < .001 | -0.44 | -0.69 | -0.18 | 0.001 | 0.39 | -0.07 | 0.86 | 0.092 |
| R5, %[†] | | | | | | | | | | | | |
| Crude | -0.70 | -0.93 | -0.48 | < .001 | -0.65 | -0.89 | -0.41 | < .001 | 0.65 | 0.33 | 0.97 | < .001 |
| Adjusted[‡] | -0.59 | -0.85 | -0.33 | < .001 | -0.48 | -0.75 | -0.21 | < .001 | 0.73 | 0.28 | 1.18 | 0.003 |
| R5-20, cmH2O/L/s | | | | | | | | | | | | |
| Crude | -0.66 | -0.90 | -0.42 | < .001 | -0.60 | -0.86 | -0.35 | < .001 | 0.68 | 0.37 | 0.99 | < .001 |
| Adjusted[‡] | -0.59 | -0.8 | -0.39 | < .001 | -0.49 | -0.71 | -0.28 | < .001 | 0.48 | 0.22 | 0.74 | < .001 |
| X5, cmH2O/L/s[†] | | | | | | | | | | | | |
| Crude | -0.60 | -0.85 | -0.35 | < .001 | -0.57 | -0.83 | -0.31 | < .001 | 0.62 | 0.29 | 0.95 | < .001 |
| Adjusted[‡] | -0.49 | -0.72 | -0.26 | < .001 | -0.45 | -0.68 | -0.22 | < .001 | 0.45 | 0.19 | 0.70 | 0.001 |
| AX, cmH2O/L[†] | | | | | | | | | | | | |
| Crude | -0.69 | -0.92 | -0.46 | < .001 | -0.77 | -0.99 | -0.56 | < .001 | 0.74 | 0.46 | 1.02 | < .001 |
| Adjusted[‡] | -0.68 | -0.85 | -0.51 | < .001 | -0.46 | -0.78 | -0.14 | 0.006 | 0.50 | 0.24 | 0.77 | < .001 |
| AX, Z-score | | | | | | | | | | | | |
| Crude | -0.71 | -0.95 | -0.47 | < .001 | -0.77 | -0.99 | -0.56 | < .001 | 0.40 | -0.02 | 0.81 | 0.061 |
| Adjusted[‡] | -0.6 | -0.88 | -0.33 | < .001 | -0.64 | -0.87 | -0.40 | < .001 | 0.34 | -0.25 | 0.92 | 0.248 |
| AX, %[†] | | | | | | | | | | | | |
| Crude | -0.80 | -1.02 | -0.57 | < .001 | -0.81 | -1.03 | -0.59 | < .001 | 0.59 | 0.17 | 1.00 | 0.008 |
| Adjusted[‡] | -0.90 | -1.2 | -0.59 | < .001 | -0.83 | -1.13 | -0.52 | < .001 | 0.59 | 0.17 | 1.00 | 0.008 |

[†] IOS parameters were log transformed to obtain normal distributions.

[‡] Variables were adjusted for age, gender.

but it has so far been best characterized for use with asthma patients. People with asthma have been shown to exhibit more negative R5, AX, Fres, and X5 compared to controls [26, 29–31]. A sensitivity cut-off in AX and R5 has been suggested to detect bronchodilator responses [25, 32, 33] and the control states of asthma [6, 27, 31]. Our study supports the possibility that AX is the most sensitive indicator of airway obstruction and resistance in BO patients specifically.

R5 Z-score showed the most statistically significant OR for airway resistance of the plethysmography in this study. In contrast to most of the IOS parameters that represented high OR values for the obstructive spirometry variables; FEV1, FEV1/FVC, and FEF25-75%, only specific IOS parameters with respect to plethysmographic resistance showed statistically significant results. These results represent the three-dimensional anatomical relationship between resistance or reactance obtained through IOS by tidal breathing, through spirometry using forced exhalation, and through body plethysmography. Raw% and sRaw% also showed good AUC in ROC analysis for the diagnosis of BO. Specific airway resistance, sRaw, can be measured plethysmographically through the simultaneous measurement of airflow and volume swing during tidal breathing and is applicable to children over two years of age. This measure provides an estimate of resistance that is practically independent of body size or age beyond infancy being the product of Raw and thoracic gas volume [34, 35]. It reflects the overall dimensions of the airway, including the effect of lung expansion on its caliber, and so it is a

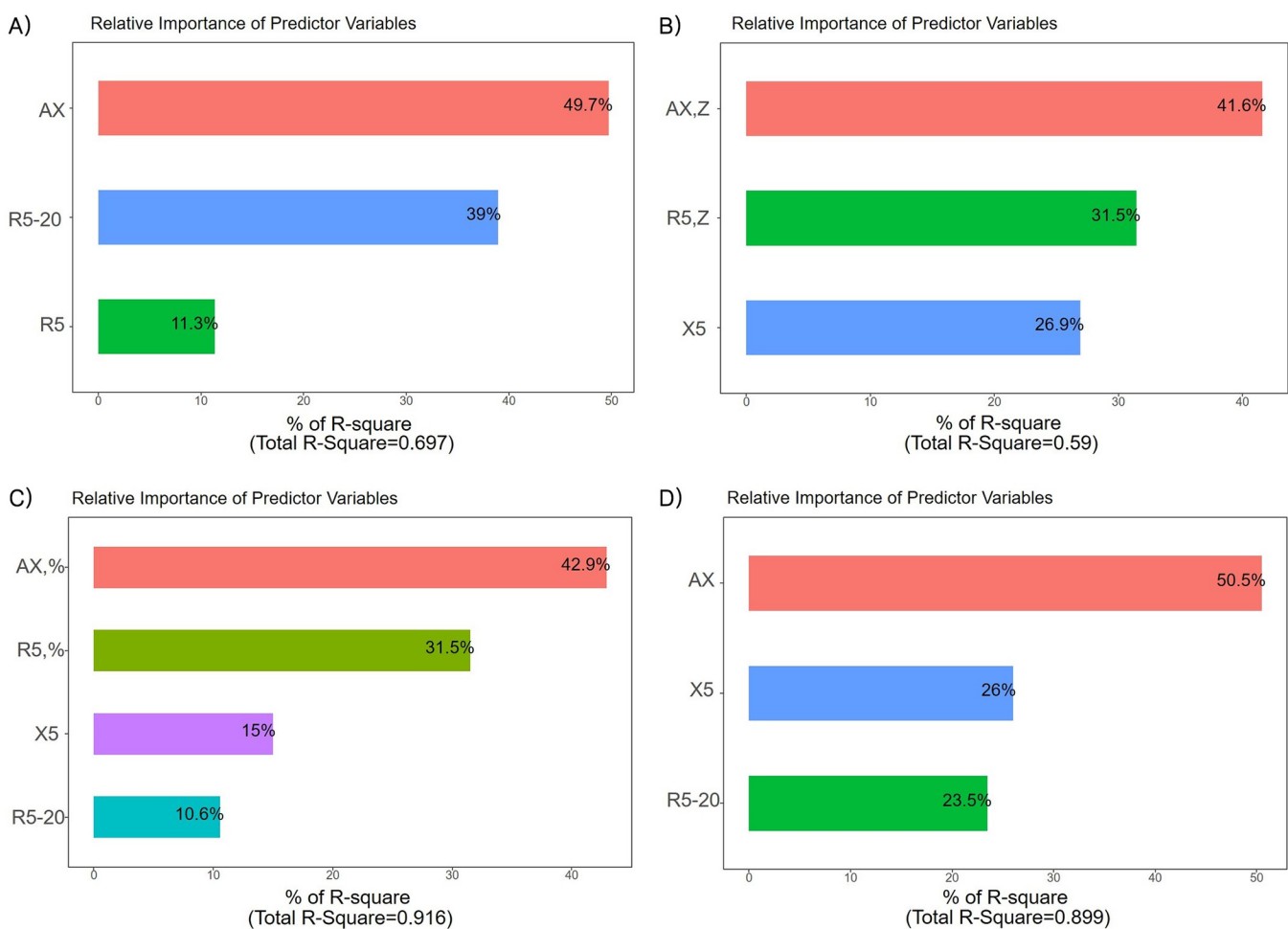

**Fig 3.** Relative importance as predictors of IOS variables for PFT parameters; A) FEV1 B) FEF25-75% C) Raw% (D) sRaw%.

more dynamic parameter compared to Raw [35, 36]. Raw itself has been reported to be more sensitive to detecting an obstruction in the extra-thoracic or large central airways than in the peripheral system [36, 37]. Our findings indicated that plethysmographic resistance showed an

**Table 3. Multivariate logistic regression for PFT parameters with IOS.**

|  | FEV1 | | | | FEF25-75 | | | | sRaw | | | |
|---|---|---|---|---|---|---|---|---|---|---|---|---|
|  | **OR** | **lcl** | **ucl** | **P** | **OR** | **lcl** | **ucl** | **P** | **OR** | **lcl** | **ucl** | **P** |
| R5, cmH2O/L/s | 2.11 | 1.36 | 3.80 | 0.004 | 2.90 | 1.64 | 6.85 | 0.002 | 2.94 | 1.48 | 8.50 | 0.012 |
| R5, Z-score | 3.94 | 1.70 | 12.75 | 0.006 | 5.96 | 2.14 | 27.31 | 0.005 | 4.85 | 1.20 | 37.86 | 0.022 |
| R5, % | 1.05 | 1.02 | 1.09 | 0.006 | 1.06 | 1.03 | 1.11 | 0.004 | 1.09 | 1.03 | 1.20 | 0.023 |
| R5-20, cmH2O/L/s | 2.55 | 1.51 | 5.14 | 0.002 | 1.17 | 1.08 | 1.26 | < 0.001 | 3.74 | 1.73 | 12.02 | 0.005 |
| X5, cmH2O/L/s | 2.53 | 1.39 | 7.32 | 0.025 | 2.61 | 1.44 | 7.54 | 0.019 | 3.30 | 1.42 | 12.29 | 0.027 |
| AX, cmH2O/L | 1.06 | 1.02 | 1.11 | 0.005 | 1.00 | 1.00 | 1.01 | < 0.001 | 1.08 | 1.03 | 1.16 | 0.018 |
| AX, Z-score | 2.84 | 1.62 | 6.74 | 0.003 | 1.13 | 1.05 | 1.22 | 0.003 | 1.59 | 0.82 | 3.61 | 0.190 |
| AX, % | 1.01 | 1.00 | 1.02 | 0.008 | 1.00 | 1.00 | 1.00 | 0.030 | 1.00 | 1.00 | 1.01 | 0.212 |

Covariates included in multivariate model: age, gender.

**Table 4. Cut off, sensitivity, specificity, and their area under the curve with 95% CI of IOS and plethysmographic parameters for the diagnosis of BO.**

|  | Cut-off | 95% CI |  | Sensitivity | Specificity | AUC |
|---|---|---|---|---|---|---|
| R5, cmH2O/L/s | 6.665 | 0.377 | 0.51 | 69.4 | 41.2 | 0.643 |
| R5, Z-score | 0.909 | 0.607 | 0.738 | 76.1 | 62.5 | 0.869 |
| R5, % | 136.15 | 0.636 | 0.759 | 65.2 | 81.2 | 0.882 |
| R5-20, cmH2O/L/s | 1.538 | 0.662 | 0.766 | 73.6 | 64.7 | 0.871 |
| X5, cmH2O/L/s | -0.048 | 0.654 | 0.762 | 66.7 | 70.6 | 0.870 |
| AX, cmH2O/L | 26.78 | 0.672 | 0.773 | 80.6 | 64.7 | 0.874 |
| AX, Z-score | 1.648 | 0.702 | 0.807 | 71.7 | 93.7 | 0.912 |
| AX, % | 131.85 | 0.707 | 0.811 | 80.4 | 56.2 | 0.915 |
| Raw % | 103.0 | 0.661 | 0.827 | 76.5 | 87.5 | 0.993 |
| sRaw % | 203.5 | 0.652 | 0.824 | 64.7 | 87.5 | 0.995 |

AUC: area under the curve; CI: confidence interval.

excellent correlation with the IOS parameters relating to peripheral airway resistance such as AX and R5.

Our study has two notable limitations: (1) The sample size was small, and the majority of the BO patients had undergone HSCT, so the presentation of different pulmonary function characteristics was limited. The BO group was heterogeneous in terms of pathophysiological status and severity at the time of IOS; (2) In those control cases where IOS was performed before HSCT, previous immunosuppressive therapy may have affected lung function. Despite these limitations, our study has the strengths of reporting IOS in children with BO, where there is otherwise insufficient research evidence, and approaching various PFTs in multiple ways to analyze the relationship from various perspectives.

In conclusion, the resistance and reactance IOS parameters were significantly worse in the BO group and were highly correlated with conventional PFTs. AX and R5 are specific parameters that can accurately detect pathological small airway obstruction in pediatric BO patients independently of the PFTs. IOS is a diagnostic method that can be easily performed and provides quantitative indicators for the identification of the severity of airway obstruction in young children patients with BO.

## Supporting information

**S1 Table. Minimal data set.** This table shows the minimal data set that was used in this study. (PDF)

## Author Contributions

**Conceptualization:** Hye Jin Lee, Jong-seo Yoon.

**Data curation:** Hye Jin Lee, Hwan Soo Kim.

**Formal analysis:** Hye Jin Lee.

**Funding acquisition:** Hye Jin Lee.

**Investigation:** Hye Jin Lee.

**Methodology:** Hye Jin Lee.

**Project administration:** Hye Jin Lee.

**Resources:** Hye Jin Lee.

**Software:** Hye Jin Lee.

**Supervision:** Hye Jin Lee.

**Validation:** Hye Jin Lee.

**Visualization:** Hye Jin Lee.

**Writing – original draft:** Hye Jin Lee.

**Writing – review & editing:** Hye Jin Lee.

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
