## [Decision Letter · Decision Letter 0]

24 Oct 2022

PONE-D-22-23545Impulse oscillometry system for the diagnosis of bronchiolitis obliterans in children; comparison with conventional pulmonary function testsPLOS ONE

Dear Dr. Lee,

Thank you for submitting your manuscript to PLOS ONE. After careful consideration, we feel that it has merit but does not fully meet PLOS ONE’s publication criteria as it currently stands. Therefore, we invite you to submit a revised version of the manuscript that addresses the points raised during the review process.

We look forward to receiving your revised manuscript.

Kind regards,

Dong Keon Yon, MD, FACAAI

Academic Editor

PLOS ONE

Journal Requirements:

   "NO authors have competing interests"

Additional Editor Comments:

Thank you for submitting your manuscript. The reviewers and I believe it is of potential value for our readers. However, the reviewers have raised a number of very important issues, and their excellent comments will need to be adequately addressed in a revision before the acceptability of your manuscript for publication in the Journal can be determined. We cannot guarantee that your revised paper will be chosen for publication; this would be solely based on how satisfactorily you have addressed the reviewer comments.

#1. The categorical and continuous variables were compared using the χ2 and Student’s t‐ test, respectively -> Please cite the statistical guideline such as DOI: https://doi.org/10.54724/lc.2022.e1

#2. IOS and PFT parameters were analyzed using Spearman's rank correlation since they did not follow a normal distribution, and we performed multivariate logistic and linear regression analysis between these two variables

-> If the normal distribution is not satisfied, linear correlation and logical regression models should not be performed.

i.e.) Yon DK, Hwang S, Lee SW, Jee HM, Sheen YH, Kim JH, Lim DH, Han MY. Indoor Exposure and Sensitization to Formaldehyde among Inner-City Children with Increased Risk for Asthma and Rhinitis. Am J Respir Crit Care Med. 2019 Aug 1;200(3):388-393. doi: 10.1164/rccm.201810-1980LE. PMID: 30958969.

TZCA, total eosinophil count, total IgE, Rrs5, and FeNO data were log transformed and Rrs5–20 data were square root transformed to obtain normal distributions.

Please transform to obtain normal distributions (using log- or square root-transformation).

#3. This is a mesmerizing paper.

#4.

Author Contributions

Conceptualization: Hye Jin Lee, Jong-seo Yoon.

Data curation: Hye Jin Lee.

Formal analysis: Hye Jin Lee.

Investigation: Hye Jin Lee, Seong Koo Kim, Jae Wook Lee, Jong-seo Yoon, Nack-Gyun Chung, Bin Cho.

-> Please add other contributors in acknowledgements.

Reviewers' comments:

Reviewer's Responses to Questions

**Comments to the Author**

1. Is the manuscript technically sound, and do the data support the conclusions?

Reviewer #1: Yes

Reviewer #2: Yes

2. Has the statistical analysis been performed appropriately and rigorously? 

Reviewer #1: Yes

Reviewer #2: Yes

3. Have the authors made all data underlying the findings in their manuscript fully available?

Reviewer #1: Yes

Reviewer #2: Yes

4. Is the manuscript presented in an intelligible fashion and written in standard English?

Reviewer #1: Yes

Reviewer #2: Yes

5. Review Comments to the Author

Reviewer #1: In his manuscript: Hye Jin Lee investigated an important issue diagnosis bronchiolitis obliterans using impulse oscillometry. It is a very interesting topic and the author should be praised for undertaking the investigation of pulmonary function assessment methods alternative to spirometry There are some issues however that in my opinion need to be addressed before publication.

1. Title – impulse oscillometry cannot be used for the diagnosis of bronchiolitis obliterans. It can be used for the assessment of pulmonary function in patients with bronchiolitis obliterans diagnosed in CT

2. There are published studies on good correlation between resistance indices of impulse oscillometry and spirometry as well as bodyplethysmography parameters. Why did the authors suspect that it might be different in bronchiolitis obliterans? This should be addressed.

3. Study population: the author state that in the control group was 1 healthy child. Yet later this healthy child is described as having Swyer James syndrome which can definitely influence the breathing mechanics. I do not think this child should be included in the control group

4. Results: Author report that “R5 values were significantly worse in the BO group” yet in table 1 no such difference is present for raw R5 values P=0.786. This leads to another issue: why does the Author present the results differently: sometimes as raw and predicted values (table 1) and sometimes as z-scores (table 2 and 3) this should be explained

5. Linear Correlation – Raw is not a spirometry parameter

6. ROC and Logistic regression: Why did the Author choose 103% for Raw and 203% for sRaw as cut-off values. If the aim of the study was to compare both IOS and bodyplethysmography parameters with spirometry it should have been stated in the aims

7. ROC and Logistic regression: Why did the author chose as the criteria of obstructive patterns FEV1, FEV1/FVC (I presume %predicted- not stated in the text) of 75 and FEF25075% of 60? The spirometric criterium of obstructive pattern of ventilation is FEV1/FVC z-score <-1.645 . The author also does not state whether just 1 or all the three criteria had to be fulfilled.

8. DISCUSSION: what does: The deterioration of the peripheral airway tissue” mean?

9. All the abbreviations used in tables should be explained

Reviewer #2: This is a nice study about the technique of IOS in BO in children which further strengthen the role and benefits in pediatric clinical practice. The results are also theoretically reasonable and conclusion is appropriate. I suggest this article is worth to read of the pediatric pulmonologists.

6. PLOS authors have the option to publish the peer review history of their article (what does this mean?). If published, this will include your full peer review and any attached files.

Reviewer #1: No

Reviewer #2: No

---

## [Author Response · Author response to Decision Letter 0]

14 Dec 2022

< Response to Reviewers >

- We sincerely appreciate the reviewers for their comments and questions, and below we address each comment and question in detail.

#1. The categorical and continuous variables were compared using the χ2 and Student’s t‐ test, respectively -> Please cite the statistical guideline such as DOI: https://doi.org/10.54724/lc.2022.e1

- The manuscript has been revised on page 5 and Table 1 as follows, according to the statistical guidelines suggested by the reviewers.

: Descriptive analyses of qualitative variables are expressed as number of patients for each category and percentage. Quantitative variables are presented as mean and standard deviation for normally distributed variables or as median and interquartile range (IQR) when not. Comparisons between patients with BO and control group were done using Chi-square test for categorical variables and Mann-Whitney test for continuous variables. If there is at least one cell that has a frequency of five or less, an additional correction was done with Fisher exact test. (page 5)

#2. IOS and PFT parameters were analyzed using Spearman's rank correlation since they did not follow a normal distribution, and we performed multivariate logistic and linear regression analysis between these two variables

-> If the normal distribution is not satisfied, linear correlation and logical regression models should not be performed.

i.e.) Yon DK, Hwang S, Lee SW, Jee HM, Sheen YH, Kim JH, Lim DH, Han MY. Indoor Exposure and Sensitization to Formaldehyde among Inner-City Children with Increased Risk for Asthma and Rhinitis. Am J Respir Crit Care Med. 2019 Aug 1;200(3):388-393. doi: 10.1164/rccm.201810-1980LE. PMID: 30958969.

TZCA, total eosinophil count, total IgE, Rrs5, and FeNO data were log transformed and Rrs5–20 data were square root transformed to obtain normal distributions.

Please transform to obtain normal distributions (using log- or square root-transformation).

- The manuscript has been revised on page 5 and Table 2 as follows, in response to the reviewers' comments:

: The IOS data were square-root transformed to obtain a normal distribution (Table 2), and a general linear model analysis between the conventional PFTs was presented with beta coefficients with 95% confidence intervals (CIs). Results of the multivariate logistic regression with obstructive PFT parameters presented as odds ratios (OR) with 95% CI. All multivariate outcomes were adjusted for confounding by age and sex (based on the assumption that these are potential confounders). (page 5)

Reviewer #1: 

1. Title – impulse oscillometry cannot be used for the diagnosis of bronchiolitis obliterans. It can be used for the assessment of pulmonary function in patients with bronchiolitis obliterans diagnosed in CT

- The title has been revised on page (1) as follows, in response to the reviewers' comments:

: “Impulse oscillometry system for assessing small airway dysfunction in pediatric bronchiolitis obliterans; association with conventional pulmonary function tests”

2. There are published studies on good correlation between resistance indices of impulse oscillometry and spirometry as well as body plethysmography parameters. Why did the authors suspect that it might be different in bronchiolitis obliterans? This should be addressed.

- The manuscript has been revised on the page (3, 6) as follows, in response to the reviewers' comments:

: The correlation between IOS and conventional PFT has already been evaluated in children with asthma [7,9-12], cystic fibrosis [13,14] and other chronic obstructive pulmonary diseases [15,16]. (page 3). To the best of our knowledge, studies of IOS with pediatric BO patients are limited. In young children with BO, irreversible fixed small airway obliteration occurs in a more severe form compared to other obstructive lung diseases. Therefore, spirometry requiring forced ventilation or the body box with long test times have a low success rate, particularly in severe cases. (page 3)

: In this study, out of 89 IOS measurements, only 43 spirometry and 29 body plethysmographic measurements were performed simultaneously, and there were cases of failure in patients who were too young or unable to perform forced breathing due to severe BO. (page 6)

: The purpose of this study is, first, to verify the correlation between IOS, spirometry, and body plethysmography in children with BO. Second, and more importantly, we tried to suggest that the IOS parameters are relatively easily obtainable, and reliable indicators even in patients with severe BO who cannot perform a conventional PFTs. (page 3)

3. Study population: the author state that in the control group was 1 healthy child. Yet later this healthy child is described as having Swyer James syndrome which can definitely influence the breathing mechanics. I do not think this child should be included in the control group

: The 15 patients included in the control group of our study were 14 patients without previous lung disease who underwent pre-transplant pulmonary function evaluation in Dep. of hemato-oncology, and 1 healthy child who underwent pulmonary function evaluation for a regular health checkup.

: The SJS patients indicated by the reviewers were included in the BO group, not the control group (Table 1). We have described this in the study population section as follows

: Patients who underwent IOS with (b) pre-transplant evaluation in Department of hemato-oncology without previous lung disease (n=14), and (c) healthy child (n=1); Both (b) and (c) were included in the control group. (page 4)

4. Results: Author report that “R5 values were significantly worse in the BO group” yet in table 1 no such difference is present for raw R5 values P=0.786. This leads to another issue: why does the Author present the results differently: sometimes as raw and predicted values (table 1) and sometimes as z-scores (table 2 and 3) this should be explained

- The authors agree that it is necessary to clarify the specific unit of the IOS parameter in the results section. The manuscript has been revised in result section on page 6 and discussion on page 11 as follows, in response to the reviewers' comments:

: Most of the IOS parameters, R5-Z, R5 %, R5-20, X5, AX, AX-Z, AX % were statistically significantly worse in the BO group (Table 1, Fig 1) (page 6)

: In the BO group, most of the IOS parameters were significantly worse except raw R5 cmH2O/L/s. As there is no general reference range of IOS specific to pediatric patients to date, we assumed that validation should include raw values, Z-scores, and % predictive values with all different units. (page 11)

5. Linear Correlation – Raw is not a spirometry parameter

- The manuscript has been revised the title of table 2 (on page 8) as follows, in response to the reviewers' comments:

: General linear regression analysis of airway resistance and reactance parameters between IOS, spirometry, and plethysmographic resistance.

6. ROC and Logistic regression: Why did the Author choose 103% for Raw and 203% for sRaw as cut-off values. If the aim of the study was to compare both IOS and body plethysmography parameters with spirometry it should have been stated in the aims

- The manuscript has been revised in the introduction section on the page (3) as follows, in response to the reviewers' comments:

: In young children with BO, irreversible fibrosis and fixed small airway obliteration occurs in a more severe form compared to other obstructive lung diseases. Therefore, spirometry requiring forced ventilation or the body box with long test times has a low success rate, particularly in severe cases. The purpose of this study is, first, to verify the correlation between IOS, spirometry, and body plethysmography in children with BO. Second, and more importantly, we tried to suggest that the IOS parameters are relatively easily obtainable, and reliable indicators even in patients with severe BO who cannot perform a conventional PFTs. (page 3) 

: The normal reference ranges for sRaw also have not been established in children, so we used 203.5% as the cut-off values following our ROC analysis in this study. (page 7)

7. ROC and Logistic regression: Why did the author chose as the criteria of obstructive patterns FEV1, FEV1/FVC (I presume %predicted- not stated in the text) of 75 and FEF25075% of 60? The spirometric criterium of obstructive pattern of ventilation is FEV1/FVC z-score <-1.645. The author also does not state whether just 1 or all the three criteria had to be fulfilled.

- The manuscript has been revised on the page (7-8) as follows, in response to the reviewers' comments:

: The cutoffs for normal values used for FEV1, FEF25-75% and sRaw were 75%, 60% and 203.5% of predicted, respectively. The obstructive group was defined when one of these criteria was met. We used a cutoff value for FEV1 < 75% based on the criteria for the definition of BO from the National Institutes of Health (NIH) consensus, 2014 [20]. FEF25-75% is a potentially sensitive parameter for evaluating small airway function in pediatrics [21]. There are no guidelines regarding normal values in children, and previous studies have suggested thresholds of 60% or 65% in children to define peripheral airflow obstruction [22,23]. 

8. DISCUSSION: what does: The deterioration of the peripheral airway tissue” mean?

- The manuscript has been revised on the page (12) as follows, to help better understand the context of the manuscript.

: Loss of compliance due to extensive fibrosis of peripheral airway tissue in BO patients can be better represented by this overall value than by resistance or reactance parameters at specific frequencies.

9. All the abbreviations used in tables should be explained

- We added the abbreviations in Tables 1-3 to help better understand the manuscript.

---

## [Decision Letter · Decision Letter 1]

27 Dec 2022

Impulse oscillometry system for assessing small airway dysfunction in pediatric bronchiolitis obliterans; association with conventional pulmonary function tests

PONE-D-22-23545R1

Dear Dr. Lee,

We’re pleased to inform you that your manuscript has been judged scientifically suitable for publication and will be formally accepted for publication once it meets all outstanding technical requirements.

Kind regards,

Dong Keon Yon, MD, FACAAI

Academic Editor

PLOS ONE

Additional Editor Comments (optional):

This is an excellent paper.

Reviewers' comments:

Reviewer's Responses to Questions

**Comments to the Author**

1. If the authors have adequately addressed your comments raised in a previous round of review and you feel that this manuscript is now acceptable for publication, you may indicate that here to bypass the “Comments to the Author” section, enter your conflict of interest statement in the “Confidential to Editor” section, and submit your "Accept" recommendation.

Reviewer #1: All comments have been addressed

2. Is the manuscript technically sound, and do the data support the conclusions?

Reviewer #1: Yes

3. Has the statistical analysis been performed appropriately and rigorously? 

Reviewer #1: Yes

4. Have the authors made all data underlying the findings in their manuscript fully available?

Reviewer #1: Yes

5. Is the manuscript presented in an intelligible fashion and written in standard English?

Reviewer #1: Yes

6. Review Comments to the Author

Reviewer #1: All comments have been addressed and the aricle can be published without any further modifications.

7. PLOS authors have the option to publish the peer review history of their article (what does this mean?). If published, this will include your full peer review and any attached files.

Reviewer #1: No

---

## [Editor Report · Acceptance letter]

30 Jan 2023

PONE-D-22-23545R1 

Impulse oscillometry system for assessing small airway dysfunction in pediatric bronchiolitis obliterans; association with conventional pulmonary function tests 

Dear Dr. Lee:

I'm pleased to inform you that your manuscript has been deemed suitable for publication in PLOS ONE. Congratulations! Your manuscript is now with our production department. 

Kind regards, 

on behalf of

Dr. Dong Keon Yon 

Academic Editor

PLOS ONE